# A Novel Ruthenium(II) Polypyridyl Complex Bearing 1,8-Naphthyridine as a High Selectivity and Sensitivity Fluorescent Chemosensor for Cu^2+^ and Fe^3+^ Ions

**DOI:** 10.3390/molecules24224032

**Published:** 2019-11-07

**Authors:** Chixian He, Shiwen Yu, Shuye Ma, Zining Liu, Lifeng Yao, Feixiang Cheng, Pinhua Liu

**Affiliations:** 1Center for Yunnan-Guizhou Plateau Chemical Functional Materials and Pollution Control, Qujing Normal University, Qujing 655011, China; cxhe@mail.qjnu.edu.cn (C.H.); LZning151@163.com (Z.L.); 2College of Chemistry and Environmental Science, Qujing Normal University, Qujing 655011, China; zlq9941@163.com (S.Y.); leoyao1982@163.com (L.Y.); lph13099867071@sina.com (P.L.); 3Department of Medicine, Qujing Qilin Vocational and Technical School, Qujing 655000, China; msy_1009@163.com

**Keywords:** Ru(II) polypyridyl complex, 1,8-naphthyridine, fluorescence, chemosensor, Cu^2+^, Fe^3+^

## Abstract

A novel ruthenium(II) polypyridyl complex bearing 1,8-naphthyridine was successfully designed and synthesized. This complex was fully characterized by EI-HRMS, NMR, and elemental analyses. The recognition properties of the complex for various metal ions were investigated. The results suggested that the complex displayed high selectivity and sensitivity for Cu^2+^ and Fe^3+^ ions with good anti-interference in the CH_3_CN/H_2_O (1:1, *v*/*v*) solution. The fluorescent chemosensor showed obvious fluorescence quenching when the Cu^2+^ and Fe^3+^ ions were added. The detection limits of Cu^2+^ and Fe^3+^ were 39.9 nmol/L and 6.68 nmol/L, respectively. This study suggested that this Ru(II) polypyridyl complex can be used as a high selectivity and sensitivity fluorescent chemosensor for Cu^2+^ and Fe^3+^ ions.

## 1. Introduction

With the development of agriculture and industry, the application of heavy and transition metals becomes more and more prevalent, which leads to these kinds of metal ions entering ecological and environmental systems. These metal ions have posed significant damage to living organisms even at per million concentrations due to their toxicity, accumulation, and low rate of clearance, especially to the health of human beings [1,2,3,4]. Among these metal ions, Fe^3+^ is the most abundant essential trace element in the organism. It is involved in several biological processes, such as oxygen transport, cellular respiration, enzymatic catalysis regulation of transcription, DNA repair [5,6,7,8], and so on. Similarly, Cu^2+^ is another indispensable element in the human body after Fe^3+^. It is required for a number of physiological activities of the organism including enzymatic catalysis, nerve conduction, and hematopoiesis. Either the deficiency or excess of Fe^3+^ and Cu^2+^ can disturb the balance of cellular systems and metabolism. For example, exposure to high levels of Fe^3+^ and Cu^2+^ can cause many diseases, such as Menkes disease, Alzheimer’s disease, Wilson disease, gastrointestinal disorders, kidney damage, hepatitis, cancer, and neurodegenerative disease [9,10,11]. Therefore, the detection and recognition of Fe^3+^ and Cu^2+^ have very important practical significance. The development of single molecular light-up probes for Fe^3+^ and Cu^2+^ has received increasing attention in recent years [12,13]. Wang et al. reported a rhodamine B hydrazine derivative that had a sufficiently satisfactory selective response to Fe^3+^ and Cu^2+^ [14]. Ajayakumar et al. designed and synthesized a new Rubp-Ptz in which the phenothiazine moiety was covalently linked to one of the bipyridine units of Ru(bpy)_3_^2+^. Excitation of Ru(bpy)_3_^2+^ led to electron transfer from the phenothiazine moiety to the metal to ligand charge transfer (MLCT ) excited state of Ru(bpy)_3_^2+^, which resulted in efficient quenching of the luminescence. The phenothiazine moiety of Rubp-Ptz can be oxidized to a stable entity by Cu^2+^. The new product is incapable of electron donation to the MLCT excited state of Ru(bpy)_3_^2+^. The emission of the Ru(bpy)_3_^2+^ moiety is restored [15]. Qian et al. synthesized five coumarin derivatives, in which two compounds exhibited selectivity towards Cu^2+^ [16]. Wang et al. reported a terthiophene-derived colorimetric and fluorescent dual-channel sensor terthiophene-phenylamine TTA, which showed a significant fluorescence “turn-on” response to Fe^3+^ and an obvious fluorescence “turn-off” response to Cu^2+^ [17]. Although a large number of luminescence probes for Fe^3+^ and Cu^2+^ have been reported, most of them have disadvantages such as low sensitivity, poor selectivity, poor water solubility, and so on. Hence, the design and development of highly selective, sensitive, and water soluble chemosensors are very important.

Ru(II) complexes and in particular Ru(II) polypyridyl complexes have received much attention due to their potential application in the field of solar energy conversion [18,19,20,21,22], photocatalysis [23,24], water-oxidation catalyst [25,26], and luminescence sensing [27,28]. Similarly, they are applied in photodynamic therapy [29] and bio-probes [30,31,32] owing to their low cytotoxicity, redox stability, and water solubility.

Significant research has focused on the derivatives of 1,8-naphthyridine due to their interesting complexation properties and application. They can be used in coordination chemistry as monodentate and bidentate ligands [33]. The derivatives of 1,8-naphthyridine have been used as molecular sensor for transition and heavy metals, carboxylic acids, and guanine [34]. 1,8-Naphthyridine derivatives are remarkable fluorescent markers for nucleic acids [35,36].

Taking advantage of the excellent luminescence properties of Ru(II) polypyridyl complexes and the coordination properties of 1,8-naphthyridine derivatives, we envision that the Ru(II) polypyridyl complexes bearing 1,8-naphthyridine derivatives can also be used as molecular probes. For the aforementioned reason, we propose to synthesize a Ru(II) polypyridyl complex containing 1,8-naphthyridine derivatives. It can be used as a luminescence probe for Fe^3+^ and Cu^2+^ ions.

## 2. Experiment

### 2.1. Materials

All the solvents and reagents (analytical grade and spectroscopic grade) were purchased from reagent companies and used without further purification. 2,7-Dimethly-1,8-naphthyridine [37,38] and *cis*-Ru(bpy)_2_Cl_2_·2H_2_O [39] were obtained according to the literature procedures.

### 2.2. Instrumentation

All of these compounds were characterized by ^1^H-NMR, MS, and elementary analysis. The ^1^H-NMR and ^13^C-NMR experiments were taken using a Mercury Plus 400 MHZ spectrometer (Bruker, Karlsruhe, Germany). The chemical shifts (δ, ppm) were acquired relative to tetramethylsilane (TMS). ESI-MS spectra were recorded by using the Bruker amaZon SL mass spectrometer (Bruker, Karlsruhe, Germany). ESI-HRMS spectra were taken on a Bruker Daltonics APEXII47e mass spectrometer (Bruker, Karlsruhe, Germany). The C, H, and N microanalyses were carried out on a Perkin-Elmer 240C analytical instrument (PerkinElmer, Norwalk, CT, USA). FTIR spectra were taken in the range of 4000–400 cm^−1^ on a Thermo Nicolet AVATAR 360 FTIR spectrometer (Thermo Fisher Scientific, Waltham, MA, USA) using KBr pellets. Electronic absorption spectra and emission spectra were taken on a Varian Cary-100 UV-Visible spectrophotometer (Varian, Palo Alto, CA, USA) and a Hitachi F-4600 fluorescence spectrophotometer (Hitachi, Tokyo, Japan), respectively, using a 10 mm path length colorimetric ware. All theoretical calculations were taken using the Gaussian 09 program package in this study (Gaussian, Wallingford, CT, USA). The structure of complex [{Ru(bpy)_2_}_2_(μ_2_-H_2_L)](PF_6_)_4_ was optimized using the Becke-3-Lee-Yang-Parr in conjunction with the 6-311+G* basis set for C, N, and H atoms and the LANL2DZ inclusion effective core potential (ECP) for the ruthenium atom [40]. All the optimized stationary points were identified as minima (zero imaginary frequencies) via the vibrational analyses at the same level.

### 2.3. Synthesis of 1,8-Naphthyridine-2,7-Dicarbaldehyde

1,8-Naphthyridine-2,7-dicarbaldehyde was synthesized with a minor change according to the literature [37,41]. A mixture of selenium dioxide (1.11 g, 10 mmol) in anhydrous 1,4-dioxane (10 mL) was heated to 110 ℃ under a nitrogen atmosphere, then 2,7-dimethly-1,8-naphthyridine was added (316 mg, 2.0 mmol). Stirring was continued for two hours and hot filtered. Chloroform (25 mL) was added to the filtrate and washed with water (3 × 10 mL). The combined organic was washed with 5% sodium bicarbonate (aq) (25 mL), dried over anhydrous Na_2_SO_4_, and concentrated in vacuo. This residue was redissolved in hot ethyl acetate (25 mL), hot filtered, giving a pale brown powder. Yield: 214 mg (58%). ^1^H-NMR (400 MHz, DMSO-*d*_6_) δ = 8.28 (d, *J* = 8.0 Hz, 2H), 8.52 (d, *J* = 8.0 Hz, 2H), 10.40 (s, 2H).

### 2.4. Synthesis of Compound L

A mixture of 5-amino-1,10-phenanthroline (585 mg, 3.0 mmol) and 1,8-naphthyridine-2,7-dicarbaldehyde (186 mg, 1.0 mmol) in anhydrous EtOH (80 mL) was heated at 80 ℃, then 10 drops of HOAc were added. The mixture was refluxed for 24 h. A suspension was obtained. The yellow precipitate was collected by filtration, washed with hot EtOH, and dried in vacuo. Yield: 463 mg (88%) of a yellow solid. ^1^H-NMR (400 MHz, DMSO-*d*_6_) δ = 7.79–7.91 (m, 6H), 8.55–8.57 (m, 2H), 8.76–8.79 (m, 6H), 8.96–8.99 (m, 2H), 9.13–9.17 (m, 4H), 9.24–9.26 (m, 2H). ESI-MS *m*/*z* = 540.65 [M + H]^+^, 562.21 [M + Na]^+^. Theoretical exact mass: *m*/*z* = 540.18 [M + H]^+^, 563.17 [M + Na]^+^. Found: C, 74.4; H, 3.8; N, 20.8. Calculated for C_34_H_20_N_8_: C, 74.5; H, 3.7; N, 20.7%.

### 2.5. Synthesis of H_2_L

L (324 mg, 0.60 mmol) was dissolved in anhydrous CHCl_3_-EtOH (200 mL, 1:1, *v*/*v*), in which NaBH_4_ (228 mg, 6.0 mmol) was added. The solvent was stirred at room temperature for 72 h, then washed with distilled water (50 mL × 3). The combined aqueous layers were extracted with CH_2_Cl_2_, then the combined organic layers were dried over anhydrous Na_2_SO_4_. The organic solvent was evaporated in vacuo, then the crude product was purified by column chromatography on SiO_2_ (eluent: *v*(CH_2_Cl_2_)/*v*(EtOH ) = 10:1), giving a yellow solid. Yield: 130 mg (40%). ^1^H-NMR (400 MHz, DMSO-*d*_6_) δ = 5.03 (s, 4H), 6.7 (s, 2H), 7.37 (s, 2H), 7.47 (dd, *J* = 4.0, 8.0 Hz, 2H), 7.73–7.78 (m, 4H), 7.95 (d, *J* = 8.0 Hz, 2H), 8.21 (d, *J* = 8.0 Hz, 2H), 8.78 (d, *J* = 4.0, 2H), 8.90 (d, *J* = 8.0 Hz, 2H), 9.16 (d, *J* = 4.0 Hz, 2H). ESI-MS *m*/*z* = 544.64 [M + H]^+^. Theoretical exact mass: *m*/*z* = 544.21 [M + H]^+^. Found: C, 75.1; H, 5.3; N,19.6. Calculated for C_34_H_24_N_8_: C, 75.0; H, 4.4; N, 20.6%.

### 2.6. Synthesis of [{Ru(bpy)_2_}_2_(μ_2_-H_2_L)](PF_6_)_4_

Under a nitrogen atmosphere, a mixture of *cis*-Ru(bpy)_2_Cl_2_·2H_2_O (207 mg, 0.40 mmol) and H_2_L (107 mg, 0.20 mmol) in 100 mL ethylene glycol in a three necked flask was heated to 150 ℃ for 12 h to give a deep red solution. The solvent was evaporated in vacuo. The crude product was purified twice by column chromatography on Al_2_O_3_, first using CH_3_CN and EtOH (8:1, *v*/*v*) as the eluant, then using EtOH to obtain the complex [{Ru(bpy)_2_}_2_(μ_2_-H_2_L)]Cl_4_. This complex was dissolved in a minimum amount of water, then the saturated aqueous NH_4_PF_6_ was added. The product was collected by vacuum filtration and recrystallized from an acetone/diethyl ether mixture, giving a deep red powder. Yield: 117 mg (30%). ^1^H-NMR (400 MHz, DMSO-*d*_6_) δ = 4.98 (d, *J* = 4.0 Hz, 4H), 6.94 (s, 2H), 7.37 (t, *J* = 8.0 Hz, 4H), 7.53–7.63 (m, 12H), 7.75 (d, *J* = 8.0 Hz, 2H), 7.80 (d, *J* = 4.0 Hz, 2H), 7.85 (d, *J* = 4.0 Hz, 2H), 7.90–7.94 (m, 2H), 8.10–8.14 (m, 6H), 8.18–8.24 (m, 6H), 8.31 (s, 2H), 8.43 (d, *J* = 8.0 Hz, 2H), 8.82–8.90 (m, 8H), 9.11 (d, *J* = 8.0 Hz, 2H). ^13^C-NMR (400 MHz, DMSO-*d*_6_) δ = 49.6, 124.4, 124.9, 125.7, 126.7, 128.3, 131.9, 132.9, 134.2, 138.2, 138.4, 138.8, 141.5, 147.2, 148.2, 151.7, 151.9, 152.0, 152.5, 157.0, 157.1, 157.3, 163.4. ESI-HRMS: *m*/*z* = 343.0735 [M − 4PF_6_]^4+^, 505.7527 [M − 3PF_6_]^3+^, 831.1112 [M – 2PF_6_]^2+^, 1087.1864 [M – PF_6_]^+^. Theoretical exact mass for C_74_H_56_F_24_N_16_P_4_Ru_2_: *m*/*z* = 343.0737 [M − 4PF_6_]^4+^, 505.7535 [M − 3PF_6_]^3+^, 831.1132 [M − 2PF_6_]^2+^, 1086.1885 [M − PF_6_]^+^. Found: C, 65.1; H, 4.0; N, 16.7. Calculated for C_74_H_56_F_24_N_16_P_4_Ru_2_: C, 64.81; H, 4.1; N, 16.34%. IR (KBr, cm^−1^) 3445br 1624m, 1528m, 1465m, 1434m, 1312w, 1241w, 1169w, 1047w, 837s, 764m, 728w, 558m.

## 3. Result and Discussion

### 3.1. Synthesis

A novel dinuclear Ru(II) polypyridyl complex containing 1,8-naphthyridine was first designed and synthesized. The target complex was prepared in four steps (Scheme 1). The starting material 2,7-methyl-1,8-naphthyridine was synthesized according to the literature [37,38]. In the first step, 2,7-methyl-1,8-naphthyridine was oxidized to dialdehyde by the selenium dioxide. The Schiff base was obtained in good yield with the reaction of dialdehyde and 5-amino-1,10-phenanthroline. The Schiff base was then reduced by sodium borohydride to give the ligand H_2_L. All the organic compounds were purified by column chromatography. *cis*-[Ru(bpy)_2_Cl_2_]·H_2_O reacted with the ligand H_2_L to give the complex [{Ru(bpy)_2_}_2_(μ_2_-H_2_L)]Cl_4_. Then, the Cl^−^ ion of [{Ru(bpy)_2_}_2_(μ_2_-H_2_L)]Cl_4_ was exchanged by the PF_6_^−^ of saturated aqueous NH_4_PF_6_ to obtain [{Ru(bpy)_2_}_2_(μ_2_-H_2_L)](PF_6_)_4_. The ^1^H-NMR, ^13^C-NMR, and MS spectra of the complex were consistent with the expected structure. 

### 3.2. Computational Studies

The simulations of the HOMO and LUMO electron density distribution in the frontier molecular orbitals of the complex are shown in Figure 1. It was observed that the LUMO and LUMO + 1 were distributed mostly on the 2,2′-bipyridine and 1,10-phenanthroline unit and that the LUMO + 2 was localized primarily on the two 2,2′-bipyridine units, but the HOMO orbital had amplitudes on the 1,10-phenanthroline unit in one of the Ru centers. The electron distribution of HOMO and LUMO provided strong evidence of the photoinduced electron transfer (PET) process in the new complexes [42,43,44]. 

### 3.3. Selectivity of Complex to Metal Ions

The selectivity of the complex to metal ions was evaluated with fluorescence experiments. The fluorescence experiments were recorded with a concentration of 10^−5^ mol/L in the presence of tetrabutyl perchlorate amine (TBAP, 0.1 mol/L) in CH_3_CN/H_2_O (1:1, *v*/*v*) at room temperature, then 5.0 equivalents of nitrate or perchlorate salts of metal ions (Ag^+^, Ca^2+^, Co^2+^, Ba^2+^, Cr^3+^, Cd^2+^, Ni^2+^, Li^+^, Na^+^, Mg^2+^, Zn^2+^, Mn^2+^, Fe^3+^, Fe^2+^, Pb^2+^, Hg^2+^, and Cu^2+^) were added to the solution, respectively. The fluorescence properties are recorded and compared in Figure 2. As shown in Figure 2, the complex showed a strong fluorescence at 612 nm. However, the fluorescence of the solution was quenched to 48 a.u. and 23 a.u., respectively, when Cu^2+^ and Fe^3+^ were added. The quenching of the fluorescence intensities with Pb^2+^ and Hg^2+^ was small. All other metal ions tested did not cause any significant change to the luminescence intensity of the solution. The phenomena of the fluorescence indicated that the complex had a selectivity for Cu^2+^ and Fe^3+^ ions. Therefore, it can be used as a “turn-off” fluorescent chemosensor for Cu^2+^ and Fe^3+^ in the CH_3_CN/H_2_O media. This dramatic quenching of the initial luminescence of RuL induced by Cu^2+^ and Fe^3+^ was due to the reverse photoinduced electron transfer (PET) from the 2,2′-bipyridine ruthenium (II) moieties to the 1,8-naphthyridine nitrogen atom because of the decrease in the electron density upon Cu^2+^ and Fe^3+^ complexation [45]. In order to further evaluate the selectivity of [{Ru(bpy)_2_}_2_(μ_2_-H_2_L)](PF_6_)_4_ for detecting Cu^2+^ and Fe^3+^, competitive experiments were carried out at room temperature. Five equivalents of Cu^2+^ and Fe^3+^ were added to the solution of the complex (CH_3_CN/H_2_O = 1:1, *v*/*v*), respectively. Then, the other metal ions (Ag^+^, Ca^2+^, Co^2+^, Ba^2+^, Cr^3+^, Cd^2+^, Ni^2+^, Li^+^, Na^+^, Mg^2+^, Zn^2+^, Mn^2+^, Fe^2+^, Pb^2+^, Hg^2+^) were added to the mixed solution, respectively (Figure 3; Figure 4). As shown in Figure 3 and Figure 4, the fluorescence of the complex with Cu^2+^ had no obvious influence when the other metal ions existed in the system. However, Ca^2+^, Cr^3+^, and Zn^2+^ had a small influence, and the other ions had no obvious influence in the Fe^3+^ ion experiments. Therefore, the complex can be used as an excellent fluorescent chemosensor for detecting Cu^2+^ and Fe^3+^ ions [14,46]. 

### 3.4. Sensitivity of Complex to Cu^2+^ and Fe^3+^ Ions

Fluorescence titration experiments for the complex with the progressive addition of Cu^2+^ and Fe^3+^ ions were carried out. As shown in Figure 5 and Figure 6, the fluorescence intensity of the complex gradually decreased at 612 nm with an increasing concentration of Cu^2+^ and Fe^3+^ ions from 0 to 1.5 equivalents. The metal ions combined with the complex [{Ru(bpy)_2_}_2_(μ_2_-H_2_L)](PF_6_)_4_, which made the electron transfer process easier than in the complex [{Ru(bpy)_2_}_2_(μ_2_-H_2_L)](PF_6_)_4_ [46,47,48]. There was a good linear relationship between the fluorescence intensity and the concentration of Cu^2+^ ions with the correlation coefficient of R = 0.9987 when the concentration of Cu^2+^ varied from 5 × 10^−7^ mol/L to 10^−5^ mol/L (Figure 7). There was a linear relationship between the fluorescence intensity and the concentration of Fe^3+^ ions with the correlation coefficient of R = 0.9501 when the concentration of Fe^3+^ varied from 6.25 × 10^−6^ mol/L to 8.0 × 10^−6^ mol/L (Figure 8). Furthermore, the equation: 3σ/k (where σ is the standard deviation of blank measurements, k is the slope between intensity vs. the concentration of ions) was used to obtain the detection limit of Cu^2+^ and Fe^3+^ [48,49]. The detection limits of Cu^2+^ and Fe^3+^ were 39.9 nmol/L and 6.68 nmol/L, which were far lower than the maximum allowable level of the WHO for drinking water. The results suggested that the complex was potentially applicable for quantitative analysis of Cu^2+^ and Fe^3+^ ions in environmental and biological systems.

### 3.5. Mode of Binding with Cu^2+^ and Fe^3+^

In order to explore the binding stoichiometry of Cu^2+^ and Fe^3+^ with [{Ru(bpy)_2_}_2_(μ_2_-H_2_L)](PF_6_)_4_, Job’s plot analyses were used to determine binding stoichiometries. The analysis results suggested that [{Ru(bpy)_2_}_2_(μ_2_-H_2_L)](PF_6_)_4_ binds with Cu^2+^ and Fe^3+^ in 3:2 and 1:1 stoichiometry ratios, respectively (Figure 9 and Figure 10). The ESI-MS experiments were further carried out by adding 2.0 equivalents of Cu^2+^ and Fe^3+^ to the acetonitrile solution of [{Ru(bpy)_2_}_2_(μ_2_-H_2_L)](PF_6_)_4_, respectively (Appendix A). The peaks at *m*/*z* = 424.33 and 646.05 were assigned to [M − 12PF_6_ − 4ClO_4_ − 6H]^10+^ and [M − 12PF_6_ − ClO_4_ − 6H]^7+^, respectively. The calculated molecular weights of [M − 12PF_6_ − 4ClO_4_ − 6H]^10+^ and [M − 12PF_6_ − ClO_4_ − 6H]^7+^ were 423.70 and 647.65, respectively. The M was [({Ru(bpy)_2_}_2_(μ_2_-H_2_L))_3_Cu_2_](PF_6_)_12_(ClO_4_)_4_ (Appendix A). The peaks at *m*/*z* = 254.40 and 335.41 were attributed to [M − 4PF_6_ − 2ClO_4_]^6+^ and [M − 3PF_6_ − 2ClO_4_]^5+^, respectively. The calculated molecular weights of [M − 4PF_6_ − 2ClO_4_]^6+^ and [M − 3PF_6_ − 2ClO_4_]^5+^ were 254.53 and 334.43. The M was [({Ru(bpy)_2_}_2_(μ_2_-H_2_L))Fe](PF_6_)_4_(ClO_4_)_3_ (Appendix A), respectively. This indicated that the ratios of [{Ru(bpy)_2_}_2_(μ_2_-H_2_L)](PF_6_)_4_ bound with Cu^2+^ and Fe^3+^ were 3:2 and 1:1, respectively. The results were consistent with Job’s plot analyses.

## 4. Conclusions

In summary, a novel fluorescent chemosensor was synthesized from 2,7-dimethyl-1,8-naphthyridine and 5-amino-1,10-phenanthroline. The ^1^H-NMR, ^13^C-NMR, and ESI-HRMS results were consistent with the expected structures. The recognition behaviors of [{Ru(bpy)_2_}_2_(μ_2_-H_2_L)](PF_6_)_4_ toward the cation were investigated. The luminescence could be almost quenched by the 5.0 equivalents of Cu^2+^ or Fe^3+^ ions. Further study showed that the complex could be used as a “turn-off” florescent chemosensor to detect the Cu^2+^ and Fe^3+^ ions. The chemosensor showed strong anti-interference, good selectivity, and high sensitivity. The detection limits of Cu^2+^ and Fe^3+^ were 39.9 nmol/L and 6.68 nmol/L, respectively. The ESI-MS experiments indicated the modes of binding to be 3:2 and 1:1, respectively. The study provided an experimental basis for the development of Cu^2+^ or Fe^3+^ ion probes that have high sensitivity and application prospects.

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
