# Peer review of "A Novel Ruthenium(II) Polypyridyl Complex Bearing 1,8-Naphthyridine as a High Selectivity and Sensitivity Fluorescent Chemosensor for Cu2+ and Fe3+ Ions"

_molecules, 2019, doi:10.3390/molecules24224032_

Round 1

Reviewer 1 Report

The article “A novel ruthenium(II) polypyridyl complex bearing 2 1,8-naphthyridine as a high selectivity and sensitivity 3 fluorescent chemsensor for Cu2+ and Fe3+ ions “ by Feixiang Cheng is well structure and is interesting. Instead, The authors should do some changes and improve the article.

pag. 2 line 71 They should add fluorescence “Hitachi F-4600  fluorescence spectrophotometer”

pag.5 TBPA may be should be write in extended formula for the first time. It is not a usual solvent for not organic chemistry specialist.

pag.5 They improve the paper if they put some number for the decreasing of fluorescence in presence of 5 equivalents of Cu2+ and Fe3+ . They have the data.

Figure 2. They should write Intensity /a.u in the y axis. In the legend is better to change fluorescence for luminescence. They should reflect the excitation wavelength.

Figure 3 and 4 show a decreasing of quenching when they add Cu2+ and specially Fe3+ with anions respect when you add only Cu2+ and Fe3+ without anions. i.e. Fluorescence of Fe3+ with Ca2+ was higher than 300 a.u and Fe3+from figure 2 show a signal of less than 100, so they could not write “have no obvious influence ….” in pag.5 line 164. Maybe they have to add a bar in figure 3 and 4 with the quenching of the Cu2+ and Fe3+ alone with the dye.

Also, they have to add in the legend of all fluorescence figures the wavelength of excitation and Intensity /a.u. in y axis.

Figure 6. They put the x axis in scientific annotation.

Figure 7. Change the number in arrow from 1 equiv from 1.6 equiva according with the legend

Figure 8 To do a right plot and fit, they have to add more points between 0.6 and 0.8. This fit does not have a right range first and last point are not appropriate for the fit according with the insert of figure 7

In figure 9 and 10. For clarify they could add a arrow in the intercept of two fit to show values of X

The authors should refer in the text the figures of SI

Finally the authors said in the text that they obtain the completely quenching at 5 equivalents and this is no in agreement with onset of figure 7 and 6.

Reviewer 2 Report

I recommend the acceptance of this manuscript, considering my points below,

All the “chemsensor” in the manuscript should be “chemosensor”. C13-NMR data should be provided in the supporting information. Line 44, “focus” should be “attention”. Line 47, “water soluble” should be “water solubility”. A space is needed at line 80. Line 126, “stating” should be “starting”. Line 154, “shown” should be “showed”. Line 215 “minimize” should be “minimized”. The size of some fonts are different in the context. Please unify them. Some other literatures related to metal ions detection should be cited. For example,

Molecules 24 (8), 1592

Inorganica chimica acta 468, 140-145

Reviewer 3 Report

The authors synthesized a novel Ru(II) polypyridyl complex (RuL) and tested its performance as a luminescent sensor for Fe(3+) and Cu(2+) ions. The assessment of the significance of the results  would require a brief overview on the structures and operation of various types of luminescent Fe(3+) and Cu(2+) molecular sensors reported in the literature. Such an overview is, however, missing from the introduction. Instead, one can read on the roles of these ions in biological systems, involving a long list of diseases which can be caused by the high levels of these ions.  

The experiments were carried out in water/AcN 1:1 mixture – presumably RuL dissolves poorly in water which is a disadvantage in potential applications.

 The explanation of the PET effect in RuL on the basis of MO calculations is obscure. Presumably, there is a misunderstanding concerning the PET, since this effect results in fluorescence quenching (see e. g. Ref. 36 in the manuscript), and the binding of the analyte leads to a turn-on signal – in contrast to the quenching of the fluorescence of RuL by Fe(3+) and Cu(2+).

The information on the products of the reactions of RuL with Fe3+ and Cu2+ is limited. One may expect to find at least some hypothetical structures.  An effort was made to analyze the products by MS. A molecular formula is presented for the two products, based on a single m/z value in their MS – this not really convincing.

I think the significance of the results described in this paper and the evaluation of the results do not reach the high standards of Molecules.

Round 2

Reviewer 1 Report

Authors did all the changes I suggested

Author Response

1.In figure 1, change LOMO to LUMO

Response1: The “LOMO” have changed to the “ LUMO”.

2.Insert your response (point 3) to referee 3 in the manuscript

Response2: The response (point 3) to referee 3 have been added to the section of selectivity of complex to metal ions in the revised manuscript.

3.In your answer to referee 3, you mention fig 1. Please insert it in the supporting information

Response3: The fig 1 have been inserted in the supporting information as Figure S 11.

4.Add concentration units in the axes of Figs 6 and 8

Response4: The concentration units in the axes of Figs 6 and 8 were “mol/L” have added in the legend of Figs 6 and 8

Reviewer 3 Report

The authors made an effort to improve their paper at some points. I think, however, that a well-based evaluation of the experimental data requires further work. I do not support the publication of this paper in Molecules.

Some main problems

The statement „high levels of Fe3+ and Cu2+ can cause many diseases, such as Menkes disease, Alzheimer’s disease, Wilson disease, gastrointestinal disorders, kidney damage, hepatitis, cancer and neurodegenerative” sounds odd. E. g. hepatitis is caused by infection. The etiology of this wide range of human diseases exceeds the scope of this manuscript.

The explanation of the fluorescence quenching in terms of PET is still completely obscure. The reverse PET upon complexation of Fe3+/Cu2+ may play a role in the quenching, this is mentioned, however, only in the Author’s response, not in the manuscript. The notations LOMO and LOMO+2 in Fig. 1 are erroneous.

The calibration curves for Cu2+ and Fe3+ look completely different, the former is close to linear in a range, the latter shows a sigmoid shape – no explanation for this difference is provided in the paper. To the sigmoid shaped fluoresc. int. vs. [Fe3+] plot a linear function is fitted in Fig. 8. The detection limit obtained from the slope of this fitted linear function is disputable. I note that no concentration units are shown on the x axes of Figs. 6 and 8.

Author Response

1.In figure 1, change LOMO to LUMO

Response1: The “LOMO” have changed to the “ LUMO”.

2.Insert your response (point 3) to referee 3 in the manuscript

Response2: The response (point 3) to referee 3 have been added to the section of selectivity of complex to metal ions in the revised manuscript.

3.In your answer to referee 3, you mention fig 1. Please insert it in the supporting information

Response3: The fig 1 have been inserted in the supporting information as Figure S 11.

4.Add concentration units in the axes of Figs 6 and 8

Response4: The concentration units in the axes of Figs 6 and 8 were “mol/L” have added in the legend of Figs 6 and 8.